



# Mixing state of black carbon and its impact on optical properties and radiative forcing over East Asia

Xiaoyan Ma[1], Hailing Jia[1], Rong Tian[1], Fangqun Yu[2], and Jiangnan Li[3]

[1]Collaborative Innovation Center on Forecast and Evaluation of Meteorological Disasters (CIC-FEMD)/Key Laboratory for
Aerosol-Cloud-Precipitation of China Meteorological Administration, Nanjing University of Information Science &
Technology, Nanjing 210044, China
[2]Atmospheric Sciences Research Center, State University of New York, Albany, NY, USA
[3]Environment and Climate Change Canada, Victoria, BC, Canada

*Correspondence to*: Xiaoyan Ma (xma@nuist.edu.cn)

**Abstract.** BC mixing state, one of essential aerosol microphysical properties modulating its optical properties and radiative forcing, and climatic and environmental effect, has often been assumed in numerical models previously. In this study, by employing a nested version of GEOS-Chem-APM model with predicted BC mixing state, we carefully examined the effect of BC mixing state on aerosol optical properties, radiative forcing, and heating rate over East Asia (EA) and particularly in

East-Central China (ECC). The modelling results show that the mass of both BC core and coating by secondary species (coated SS) are significantly high in ECC and India due to strong anthropogenic emissions but quite low in the other regions. The ratio of total mass (BC + coated SS) to the BC over ECC can be doubled compared to pure BC core mass, indicating quite large coating over the regions with high anthropogenic emissions. Absorptive aerosol optical depth (AAOD) enhances ~40% over ECC once core-shell mixing, rather than external mixing, is taken into account, despite AOD only changes

slightly, and the resulting absorption amplification (Aa) normally varies from 1.1 to 1.8 over ECC. Aerosol direct radiative forcing (DRF) under clear sky and all sky indicate that DRF at top of atmosphere (DRF$_{TOA}$) become weaker but DRF at surface (DRF$_{SRF}$) become stronger when core-shell mixing, instead of external mixing, is taken in to account in the simulation. The simulation with core-shell mixing presented that over ECC, the regional-averaged atmospheric absorption enhances 39% while DRF$_{TOA}$ and DRF$_{SRF}$ are reported as -8.3 and -17.8 W m$^{-2}$, compared to -9.4 and -16.2 W m$^{-2}$ in the

simulation with external mixing state. The heating rate in entire column enhances with core-shell mixing state due to absorption amplification, compared to external mixing state. The heating rate at surface over ECC increases about 43%, i.e. increase from 0.7 K day$^{-1}$ (external) to 0.9 K day$^{-1}$ (core-shell) during the study period. The results in three cities (Beijing, Nanjing, Chengdu), representative of North China Plain (NCP), Yangtze River Delta (YRD), and Sichuan Basin (SCB), major anthropogenic aerosol emissions regions over EA, are also discussed in the study.




## 1 Introduction

Black carbon (BC) particles, which emitted from both anthropogenic (incomplete fossil fuel combustion) and natural sources

(biomass burning), are one of major aerosol components in atmosphere and have significant impacts on climate change and air

pollution through modifying solar radiation flux in Earth-atmosphere system (Hansen et al., 2000; Chameides and Bergin,

2002; Menon et al., 2002; Bond and Bergstrom, 2006; Ramanathan and Carmichael, 2008; Gustafsson and Ramanathan, 2016).

It is well known that direct radiative forcing (DRF) of aerosol strongly depends on its microphysical properties, including

chemical component, size distribution, morphology, and mixing state. BC particles are normally irregularly shaped and mostly

solid once emitted, and then externally or internally mixed with other aerosol types by aging process (Bond et al., 2006; Koch

et al., 2009; Peng et al., 2016; Liu et al., 2017). The coating of BC particles with secondary aerosol constituents (such as

sulfate, nitrate, ammonia, and organics) through aging process can enhance the mass absorption cross-section due to the

lensing effect, i.e. the coating materials act as a lens to focus more photons on BC particles (Fuller et al., 1999; Jacobson, 2000;

2001; Lesins et al., 2002; Bond and Bergstrom, 2006; Cheng et al., 2006; Ramanathan and Carmichael, 2008; Lack and Cappa,

2010). This is so-called light amplification (Bond and Bergstrom, 2006). The change of light absorption in the atmosphere will

modify radiation flux in the earth-atmosphere system, heating rate, as well as the atmospheric stability (Jacobson,2000; Ding et

al.,2016; Wilcox et al., 2016; Gao et al., 2018; Wang et al.,2018).

Light absorption of BC particles is quite complex since it changes with the morphology and mixing state of the

BC-containing particles (Schnaiter et al., 2005; Zhang et al., 2008; Knox et al., 2009; Peng et al., 2016; Bai et al., 2018; Liu et

al., 2018), i.e. the degree of internal mixing between BC and other particle species (i.e., non-BC components) is strongly

dependent on the atmospheric aging process, such as condensation, coagulation and heterogeneous oxidation. Previous

theoretical (Jacobson, 2001; Moffet and Prather, 2009; Zhang et al., 2016) and observational (Knox et al., 2009; Cappa et al.,

2012; Chung et al., 2012; Dong et al., 2015; Peng et al., 2016) studies showed a rather broad range of absorption enhancements

(1.05–3.05) of BC-containing particles during the atmospheric aging process. In terms of individual BC containing particles,

more coating materials result in stronger light absorption capability. The variation of BC optical properties will probably exert

significant influence on its radiative forcing based on previous studies, e.g. the studies by Jacobson's (2000, 2001) found that

BC forcing with the core-shell treatment is 50% higher than forcing obtained with the externally-mixed treatment.

During the past few decades, aerosol concentrations (including BC) over China have increased dramatically due to a

significant enhancement of anthropogenic emissions with rapid economic development, and have consequently led to serious

environmental and climatic problems (Gu et al., 2010; Zhuang et al.,2010; Wang et al., 2012; Zhuang et al., 2014; Ding et al.,

2016; Ma and Jia, 2016; Ma et al., 2018; Zhang et al.,2018). For example, the weekly average mass concentration of BC was

reported as 4.3μg m$^{-3}$ according to nine years' measurements during 2005 to 2013 in Beijing (Chen et al., 2016). Other

measurements in the summer of 2005 gave the average mass concentration of BC as 2.37 (±1.79) and 5.47 (±4.00) μg m$^{-3}$ in

Beijing and Shanghai (Zhou et al., 2009). The measurements at Mt. Tai and nearby urban area in North China indicated that

non-BC coatings could enhance the mass absorption cross-section (MAC) of BC particles by a factor of 2, i.e. light absorption

of BC is doubled (Bai et al., 2018). Such profoundly enhanced BC absorption induced by high BC loadings and increased

coating can significantly modify radiation heating of the atmosphere and thus exhibit climatic and environmental effect. The

study by Ding et al. (2016) showed that BC induces heating in the planetary boundary layer (PBL), particularly in the upper

PBL, and the resulting decreased surface heat flux substantially depresses the development of PBL and consequently enhances

the occurrences of extreme haze pollution episodes.

As our previous studies (Yu et al., 2012; Ma et al. 2012) presented, the GEOS-Chem-APM model can predict the mixing

state of BC particles, i.e. core-shell mixing state, rather than assuming either external mixing or internal mixing, as most

previous studies have done. Thus, the predicted mixing state of aerosol particles is physically more realistic than the

assumption. In this study, a nested version of GEOS-Chem-APM model is employed to investigate the mixing state of BC on

aerosol optical properties, radiative forcing, and heating rate over East Asia. The paper is organized as follows: the descriptions

of model and emissions (both anthropogenic and natural) are presented in Section 2 and 3, the simulated mass concentrations

of BC and coating and comparisons with the measurements, while aerosol optical depth and absorption amplification, radiative

forcing, and heating rate are presented in Section 4. A summary and discussion is summarized in Section 5.

## 2 Model description and experiments design

### 2.1 Nested GEOS-Chem-APM

The GEOS-Chem is a global chemistry model driven by assimilated meteorological observations from the Goddard Earth

Observing System (GEOS) of the NASA Global Modeling Assimilation Office (GMAO), including a number of

state-of-the-art modules representing various chemical and aerosol processes (e.g. Bey et al., 2001; Martin et al., 2003; Park et

al., 2004; Evans and Jacob, 2005; Liao et al., 2007; Fountoukis and Nenes, 2007) with up-to-date key emission inventories (e.g.

Guenther et al., 2006; Bond et al., 2007). The APM (Advanced Particle Microphysics) model (Yu and Luo, 2009), which has

been incorporated into GEOS-Chem, is an advanced multi-type, multicomponent, size-resolved microphysics model, in which

comprehensive microphysical processes have been parameterized, including nucleation, condensation/evaporation,

coagulation, thermodynamic equilibrium with local humidity, and dry and wet deposition. New particle formation is

parameterized with the ion-mediated nucleation mechanism (Yu, 2010). As a bin-based aerosol module (e.g. APM) is very

computational expensive, a number of algorithms have been employed to improve the computing efficiency (Yu and Luo,

2009). Prognostic aerosol compositions include secondary particles (SP, containing sulfate, ammonia, nitrate and SOAs), BC,

primary organic carbon (POC), sea salt and mineral dust. The contributions of nitrate, ammonium, and SOAs to secondary

particle growth are considered. The model has been validated extensively with a large number of relevant surface





measurements and satellite retrievals (Yu and Luo, 2009; Yu et al., 2010; Yu, 2011; Ma et al., 2012; Ma and Yu, 2014; 2015).

Global aerosol distributions and mass concentrations, optical properties, radiative forcing, as well as its uncertainties on microphysical properties of aerosol have also been examined in our previous study (Ma et al., 2012). The GEOS-Chem-APM model has been designed to explicitly predict the coating of secondary species on primary particles (sea salt, BC, POC, and dust) and spatiotemporal variations in the degree of particle mixing, thus it is possible for us to explore the impact of mixing state (external and core-shell mixing) on optical properties of BC, radiative forcing, and heating rate.

The nested-grid GEOS-Chem (version 9-01-01) with a spatial resolution of $0.5\,^\circ \times 0.667\,^\circ$ (Chen et al., 2009) is used in this study, with chemical boundary conditions provided by the GEOS-Chem global simulation at a resolution of $4\,^\circ \times 5\,^\circ$. The nested-grid domain is 70°E–150°E, 11°S–55°N. The nested-grid GEOS-Chem CTM has been evaluated and applied to the analysis of SNA (Sulfate-nitrate-ammonium) and black carbon over China (Wang et al., 2013; Kharol et al., 2013).

To examine the effect of mixing state on optical properties and DRF due to anthropogenic aerosols, two experiments are
conducted, i.e. COAT and NoCOAT, with COAT as a base experiment in which core-shell (coating) mixing state is applied, while the experiment NoCOAT is based on the base experiment except assuming aerosol particles as external mixed, i.e. no coating is taken into accounted. Both simulations are conducted for January 2014 over East Asia.

## 3 Anthropogenic and natural emissions

Carbonaceous aerosol emissions include fossil fuel and bio-fuel combustion and biomass burning. In this study,
anthropogenic carbonaceous emissions used Bond et al. (2004)'s fossil fuel and bio-fuel inventories, while biomass burning emission is based on Global Fire Emissions Database Version4 (GFED4) monthly open fire inventory (van der Werf et al., 2006). The MIX v1.1 is used to replace the Asian emission inventory (Li et al., 2017). The monthly anthropogenic emissions of $NO_x$, $SO_2$, and $NH_3$ over China are taken from the MIX emission inventory for the year of 2010 (Li et al., 2017). Carbonaceous aerosol emissions include fossil fuel and bio-fuel combustion and biomass burning. Carbonaceous aerosols in
GEOS-Chem-APM are grouped into hydrophilic and hydrophobic species. Conversion of hydrophobic to hydrophilic carbonaceous aerosols takes place with an e-folding time of 1.2 days based on Cooke et al. (1999).

The monthly mean BC emissions from anthropogenic and biomass burning over East China are shown in Fig. 1. It is obvious that anthropogenic emissions are dominant, particularly over East and Central China (ECC) (the domain is shown as a small blue square in Fig.3a). Anthropogenic emission over ECC varies between $10^{-12}$ to $10^{-10}$ kg m$^{-2}$s$^{-1}$ due to large
industrial emissions, while the emissions over other regions are normally less than $10^{-12}$ kg m$^{-2}$s$^{-1}$. High anthropogenic emissions are also found over India, with the regional-averaged anthropogenic emissions of $1.04 \times 10^{-11}$ kg m$^{-2}$s$^{-1}$, only next to ECC ($2.13e^{-11}$ kg m$^{-2}$s$^{-1}$). In contrast, the emissions from biomass burning are overall lower than $5e^{-12}$ kg m$^{-2}$s$^{-1}$ over most areas. However, biomass burning emissions over Yangtze River Delta (YRD) and Pearl River Delta (PRD) are significantly



higher than other areas, with the highest emissions over $10^{-10}$ kg m$^{-2}$s$^{-1}$, which is possibly due to corn draw burning in this

season.

**4 Black carbon concentration, and comparisons with observations**

**4.1 Black carbon (BC) burden and surface concentration**

Our earlier study (Ma et al., 2012) showed that secondary species (SS) coated on primary particles are generally much

lower than those remaining in SP, but a large fraction (up to 50~80 %) (Yu et al., 2012) can become coated on various primary

particles in certain regions, e.g. East Asia (EA). The high-resolution simulations with a nested model version over East Asia

(Fig. 2a) show that BC burden present a maximum over ECC and India, with the magnitudes of higher than 1 mg/m$^2$, but varies

between 0.1 to 1 mg/m$^2$ over most other areas. This is quite consistent with the spatial distributions of anthropogenic emissions

of BC. The region-averaged BC burden over Each Asia and ECC summarized in Table 1 indicates that the burden over ECC is

nearly 5-fold high than over EA.

Similarly, BC concentrations at surface also locate mainly over ECC, where the monthly mean BC concentrations varies

between 1.0 to 10 μg m$^{-3}$, with the average of 5.0 μg m$^{-3}$, which is nearly 5 time higher compared to EA (1.1 μg m$^{-3}$) (Fig.2d).

High coating concentrations are consistently found over these regions (Fig.2e), with the largest coating in Beijing-Tian-Hebei

(BTH), Yangtze River Delta (YRD), Sichuan Basin (SCB), and Central China (CC), corresponding to heavy haze regions in

China. The monthly mean mass of SS coating over ECC, 1.45 μg m$^{-3}$, increase by over three-fold relative to the regional mean

(0.39 μg m$^{-3}$). The ratio of total mass (BC core + coated SS) to the mass of BC core (Fig.2f), defined as $M_R$, generally ranges

from 1 to 2 over mainland China. It is noted that $M_R$ exhibits higher values in southern China (greater than 1.25) than northern

China (less than 1.25).

Fig.3a shows our model domain over EA while the small square inside domain represents ECC region. Fig.3b presents

the PDF distribution of $M_R$ for the total column, below 2km and below 1km over ECC. Over 71%, 77%, and 90% of $M_R$ varies

from 1.0 to 3.0 for both entire column, below 2km, and below 1km, respectively. The maximum of $M_R$ near surface tend to be

lower than that in entire column, and the probability of small $M_R$ increases in the case of below 1 km than two other cases,

implying that the coating on BC particles by other soluble aerosol enhances with the height. It is probably suggested that BC

particles become coated due to aging process compared to pure BC particles once emitted. According to measurements-based

studies (e.g. Peng et al., 2016), BC aging has two distinct stages, i.e. initial transformation from a fractal to spherical

morphology and subsequent growth of fully compact particles. The optical properties of BC change little during the first stage

but absorption largely enhances during the second stage. The measured timescales to achieve complete morphology

modification and an absorption amplification factor of 2.4 for BC particles are estimated to be 2.3 h and 4.6 h in Beijing. We





are unable to examine the timescales of BC morphology based on daily mean model output in current study, but it is clearly to see less coating near surface and more coating with height.

**4.2 BC aerosol optical depth (AOD), absorption AOD, and absorption amplification**

Aerosol optical properties in this study, including extinction efficiency, single scattering albedo, and asymmetry parameter, is calculated online using a computationally efficient scheme based on lookup tables (Yu et al., 2012; Ma et al, 2012). The predicted aerosol microphysical properties such as size distribution, chemical component, the coating of primary particles by volatile species, and hygroscopic growth, etc, are provided as input to Mie calculations based on core-shell Mie model developed by Ackerman and Toon (1981). The calculated aerosol optical properties depend on wavelength, core diameter, shell diameter, and refractive index. The refractive indices for sulfate, ammonia, nitrate, SOA, POC, and water are according to the corresponding values given in Aouizerats et al. (2010). The refractive indices for sea salt, BC, and dust are based on the values recommended by Krekov (1993), Bond and Bergstrom (2006), and Balkanski et al. (2007). The global aerosol optical properties predicted by GEOS-Chem-APM have been extensively evaluated against AERONET, MODIS, MISR, and SeaWiFs measurements (Ma et al., 2012; Yu et al., 2012; Ma et al., 2013; Ma and Yu, 2014; 2015).

The monthly mean aerosol optical depth (AOD) and absorptive AOD (AAOD) at a wavelength of 550 nm simulated from the experiments COAT and NoCOAT are shown in Figure 4. AOD is overall quite large over ECC, with the values over 0.7 in most regions and the maxima of 2.0 over SCB and CC. Over adjacent oceans, AODs are around 0.4~0.7 due to transport of high aerosol concentration from strong anthropogenic emission regions. The differences of AOD in two simulations (Fig.4c) indicate that largest differences are less than 0.1, i.e. less than 10%, implying little impact of mixing state on total AOD. In contrast, a significant difference in AAOD from two simulations is presented, particularly over ECC where is characterized by both high AOD and AAOD. Fig.4f and Table 1 show large differences of AAOD (Fig.4f) over ECC, i.e. 0.026 from the simulation without coating, which increase to 0.037 in the simulation with core-shell mixing. It is expected that this enhancement of absorption would influence both radiation flux and heating rate in the atmosphere, which will be presented below.

Time evolution of the simulated AOD from two experiments have been compared with the Aerosol Robotic NETwork (AERONET) retrievals (Holben et al., 1998; 2001) at the observational sites within the study domain. There are only 7 sites after we applied the cloud-screened and quality-assured AERONET Level 2.0 data (Smirnov et al., 2000), and AOD at 550 nm obtained by spectral interpolation is employed for comparisons. For consistence, the modelled monthly mean AOD in the grid cells at which the AERONET sites located, are selected to compare with the AERONET monthly results. The AODs from two experiments at all sites have little differences (Fig.5) and all agree overall well with AERONET observations in terms of both magnitudes and temporal variations. Data availability of AERONET AAOD measurements, however, is quite limited compared with AOD after the procedure of data filtering and processing as mentioned above (Fig. 6). From the limited



measurements, it is found that the model generally reproduces the variations of AAOD in Beijing and Chen-Kung-University,
but profoundly underestimates the measured values in Kanpur, India (up to 10 times lower). It is possibly because the BC
emissions over India in emission inventory are too low and have not been updated, while the emissions in China have been
updated as described earlier. The values of AAOD in the experiment COAT (take coating into account) are apparently higher
than that in the experiment NoCOAT (without considering coating), and also more close to the AERONET measurements. The
statistics summarized in Table 2 (AERONET's results not included due to very limited measurements) shows that AAOD in
COAT enhanced by roughly 30% compared to AAOD in NoCOAT.

The ratio of AAOD in COAT to that in NoCOAT is considered as absorption amplification (Aa) (Bond et al, 2006),
which is induced by the amount of both BC-core and coated SS. Figure 7 indicates the absorption amplification (Aa) and its
dependency on $M_R$ and BC core over ECC, in three representative cities in China including Beijing (BJ), Nanjing (NJ), and
Chengdu (CD), representative of North China Plain (NCP), Yangtze River Delta (YRD), and Sichuan Basin (SCB),
respectively. For the regional-averaged over ECC, the Aa values enhance with the increase of $M_R$, with rapid growth initially
($M_R < 2.0$, i.e. relatively less coating) and then slow down, especially when $M_R$ is greater than around 3.0 (more coating). In
addition, it is noted that for a fixed $M_R$, the Aa values tend to be larger for a higher BC core burden. These findings are
consistent with the results from theoretical calculations based on core-shell Mie theory (Ackerman and Toon, 1981; Seinfeld
and Pandis, 1998). The comparisons of the Aa values and its dependency on $M_R$ and BC burden in BJ, NJ and CD exhibit
some quite differences. It is shown that $M_R$ in BJ varies between 1.0 to 1.8, while $M_R$ in NJ and CD ranges from 1.1 to 3.0,
i.e. $M_R$ in BJ is overall significantly lower than in NJ and CD, indicating less coating in NCP than two other cities.
Compared to NJ and CD, steeper variation of Aa versus $M_R$ in BJ indicate that the dependency of Aa on $M_R$ is more
dramatic, especially for higher BC core burden.

### 4.3 Impact of mixing state on aerosol radiative forcing

Aerosol direct radiative forcing (DRF) is strongly associated with particle size, component, and mixing state. The mixing
state of aerosol and its impact on DRF have been explored by previous studies (Chylek et al., 1995; Lesins et al., 2002), which
found that absorption of internally mixed BC is amplified compared to the externally mixed case. Experiments in aerosol
chamber also measured this amplification (Schnaiter et al., 2005). The simulations by Jacobson (2000, 2001) found that BC
forcing with the core-shell treatment is 50% higher than forcing obtained with the externally-mixed treatment, and it was
suggested that the real forcing by BC probably fell between that from an external mixture and that from a coated core. Our
earlier study (Ma et al., 2012) based on global scale indicated that the DRF could be quite different between core-shell mixed
and externally-mixed aerosol. Compared with the experiment for externally-mixed state, the simulations with the
model-predicted mixing state (core-shell) suggested that global mean absorption increases 0.29 W m$^{-2}$, while the DRF at top of





atmosphere (TOA) increases 0.20 W m$^{-2}$. In addition, the major differences occur in northern hemisphere, specifically over

East Asia, central Africa, eastern United States and Polar Regions, i.e. source areas and its transport regions.

In this study, the simulations based on a nested GEOS-Chem-APM model version over East Asia are shown in Fig. 8.

Over clear sky condition, strong cooling occurs over East China and its adjacent ocean, as well as India, presumably due to

strong anthropogenic emissions and transport, while cooling over other regions are quite weak. The monthly mean statistics

over the entire model domain (EA) and East Central China (ECC) summarized in Table 1demonstrate that aerosol DRF over

ECC is -13.8 and -24.5 W m$^{-2}$ at TOA and surface (SFC), respectively, which is 2.1 and 2.6 times higher than over the entire

domain (-6.8 and -9.4 W m$^{-2}$). Over all sky condition (Fig.9 and Table 1), aerosol DRF decreases compared to clear sky

conditions for both TOA and the surface due to dominant cloud effect. The monthly mean DRF at TOA over all sky decrease to

-4.0 W m$^{-2}$ and -8.3 W m$^{-2}$ at TOA, and -6.4 and -17.8 W m$^{-2}$ at surface, for the entire domain and ECC, respectively. The

absorption in atmosphere by absorbing aerosols (mainly BC) thus contributes to 2.4 and 9.5 W m$^{-2}$ for the entire domain and

ECC.

The results mentioned above are based on the simulations in the case of core-shell mixing. Our study (Figure 9 and Table 1)

also show that in comparison to the simulation with external mixed aerosol (experiment NoCOAT, middle column), the

simulation with core-shell configuration (experiment COAT, left column) exhibits stronger absorption (right column), i.e. 0.7

W m$^{-2}$ more over EA and 2.7 W m$^{-2}$ more over EC. This induces a stronger cooling at surface but weaker cooling at TOA, with

the differences of 0.4 W m$^{-2}$ for EA and 1.7 W m$^{-2}$ for EC.

### 4.4 Heating rate

The radiative heating rate is equal to divergence of the net solar flux (downward minus upward) for a specific atmospheric

layer, thus the heating rate of an atmospheric layer can be computed from the ratio of net heat flux and the thickness of the

atmospheric layer (Liou, 2002). The heating rate from the ground to 10 km are presented in Figure 10 for ECC and three

representative cities in China, i.e. Beijing, Nanjing, and Chengdu. The mass concentrations of BC core and the coating by

secondary particles (SP), as well as the values of $M_R$ are also presented for analysis. Additional simulation without aerosol

effect is included as well for comparison. It is shown that the heating rating in the simulations without including the radiative

effect of aerosols is around 0.4 K day$^{-1}$ at surface and decreases with height. The radiative effect of aerosol particles

significantly enhances the heating rate below 3 km, which is consistent with the vertical profiles of aerosol mass concentrations.

The enhancement of the effect of core-shell configuration on heating rate, compared with the case of external mixing state, is

also profound, specifically below 4 km. The regional-averaged heating rate at surface over ECC enhances from 0.7 K day$^{-1}$

(external mixing) to 0.9 K day$^{-1}$ (core-shell mixing). The degree of enhancement is sensitive to mass concentrations of both

BC core and coated-SP, as well as their ratios ($M_R$). Comparisons of the results in three cities (Fig. 10b, c, d) show that in

Chengdu, not only the heating rate by aerosol particles is higher than other cities, but also the enhancement is higher (about





30%), i.e. from 1.0 (aerosol external mixing) to 1.3 (core-shell) K day$^{-1}$, which is induced by high mass concentrations of both BC core and shell. It is also noted that the $M_R$ is relatively low in Beijing in comparison to other cities, i.e. less coating mass concentration, thus the absorption amplification and the differences in heating rate between external and core-shell mixing are smaller than in Nanjing.

**5 Summary and conclusions**

BC mixing state is a key factor to influence aerosol optical properties, and its direct and indirect effect. As a major industrial region, East Asia is characterized by quite complex aerosol microphysical properties, including chemical component, size distribution, and mixing state. Many previous modelling studies have investigated the impact of mixing state on aerosol optical properties and radiative forcing, but most of them were conducted under the assumption of either

external or internal mixing. In this study, we examined the effect of BC mixing state on aerosol optical properties BC, radiative forcing, and heating rate over East Asia (EA) and particularly in East-Central China (ECC), based on a nested version of GEOS-Chem-APM model simulations, in which aerosol mixing state is predicted on line.

Two simulations are conducted during January 2014, with COAT as a base experiment in which core-shell mixing state is applied, while the experiment NoCOAT is same as COAT except external mixed is assumed. It is found that the mass of

both BC core and coating by secondary species are quite high over ECC and India, which is clearly induced by strong anthropogenic emissions. The ratio of total mass (BC core + coated SS) to the BC core, defined as $M_R$ in the study, varies from 1 to 2 over ECC, indicating that total mass could be doubled than pure BC core due to the contribution of coating in some cases. The PDF distribution of $M_R$ presents that small $M_R$ occurs more frequently below 1 km than below 2 km and entire column, implying that the coating enhances with height due to aging process. Comparisons in two experiments

indicate that AAOD over ECC increases over 40% when core-shell mixing is taken into account although total AOD remain almost identical. Time evolution of AOD and AAOD in AERONET sites within the study domain further confirm that two experiments can both capture the observed variations and magnitudes of AOD, but the modelled AAOD with COAT is obviously higher than NoCOAT and more consistent with AERONET. Absorption amplification (Aa), a function of both BC core mass and $M_R$, increases with BC mass and $M_R$ and varies normally between 1.1 to 1.8 over ECC. A comparative

analysis in three representative cities over ECC apparently show that $M_R$ and BC mass are overall lower in Beijing than in Nanjing and Chengdu, indicating relatively weak coating in Beijing. Aerosol direct radiative forcing (DRF) under clear sky from two experiments are calculated as -6.8 W m$^{-2}$ (COAT) and -7.2 W m$^{-2}$ (NoCOAT) at TOA, and -9.4 W m$^{-2}$ (COAT) and -9.0 W m$^{-2}$ (NoCOAT) at surface over the entire study domain, i.e. the simulation with core-shell mixing will reduce DRF at TOA but enhance DRF at surface. The reduction at TOA and enhancement at surface will increase to 1.1 and 1.8 W m$^{-2}$ at

TOA and surface, respectively, over ECC. The results under all sky indicate that total absorption in the atmosphere enhance



from 6.8 W m$^{-2}$ (external mixing) to 9.5 W m$^{-2}$ (core-shell) over ECC. The heating rate in the entire column increases in the simulation with core-shell mixing than with external mixing due to absorption amplification. The heating rate at surface induced by aerosols is estimated to 0.32 K day$^{-1}$, in comparison to the values of 0.4 K day$^{-1}$ (without aerosols), which further increases to 0.5 K day$^{-1}$, once core-shell mixing is considered. The results at three representative cities in ECC indicate that

both the heating rate induced by aerosol and the discrepancies between external or core-shell mixing are relatively small in Beijing, but large in Chengdu.

It is expected the change of radiation flux in atmospheric vertical profile induced by aerosol mixing state will inevitable modulate boundary layer height and consequently enhances the occurrences of haze episodes. In addition, the change of atmospheric vertical profile, as well as the change of aerosol size and hygroscopic properties and thus aerosol-cloud

interaction, will probably substantially influence surface temperature and precipitation. Further studies employing a weather or climate model are necessary to address these issues.

*Competing interests.* The authors declare that they have no conflict of interest.

*Acknowledgements.* This study is supported by the National Natural Science Foundation of China grants (41975002,41675004 and 41475005).

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





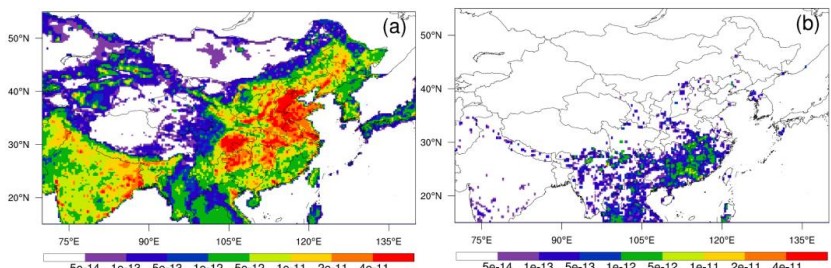

Fig.1. (a) Emissions over East Asia in this study, in which anthropogenic emissions from MIX (a), while biomass
burning emissions from GFED4 (b). Units: kg m$^{-2}$s$^{-1}$.

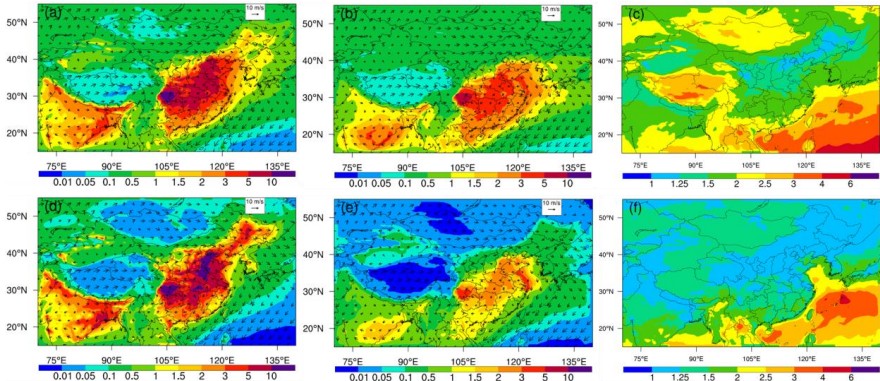

Fig. 2. The top column and bottomn column is for mass burden and surface concentration (the lowest model level).
From left to right is the mass of BC core (a,d), coated SS (b,e), and the ratio of total mass (BC core + coated SS) to the
mass of BC core ($M_R$) (c,f).    Wind fields are also shown here for analysis. Units: mg m$^{-2}$ for burden and  μ g m$^{-3}$for
surface concentration.

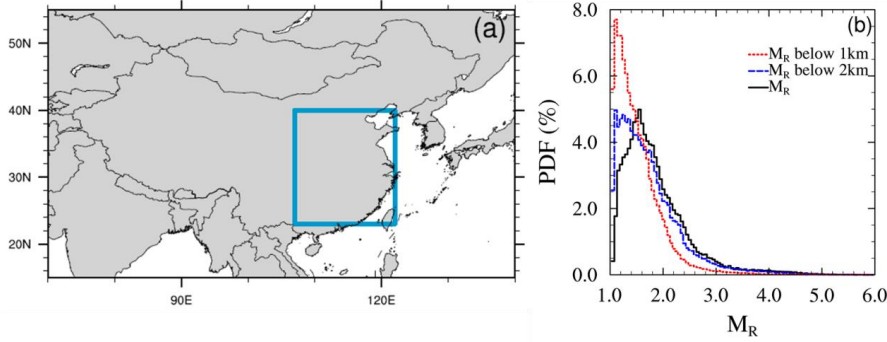

Fig.3. The model domain (a) and probability density function (PDF) distribution of $M_R$ for the entire column, below
1km, and below 2km over ECC (denoted as small square inside the model domain) (b).





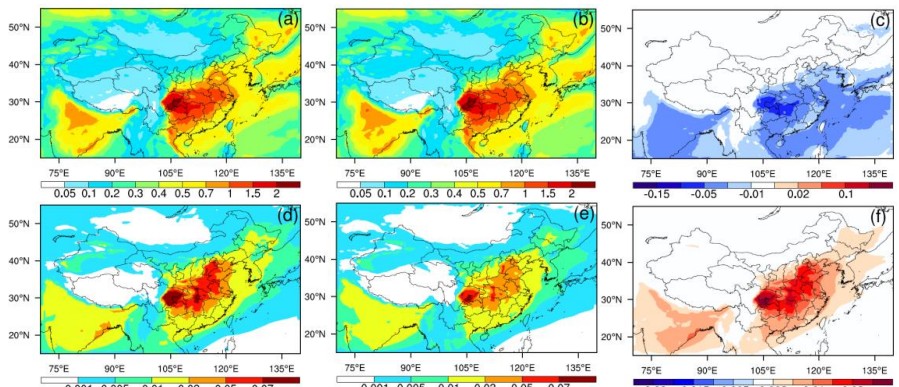

Fig. 4. Simulated AOD (upper), AAOD (lower) from the experiments COAT (left column) and NoCOAT (middle column), and their difference between two experiments (right column).


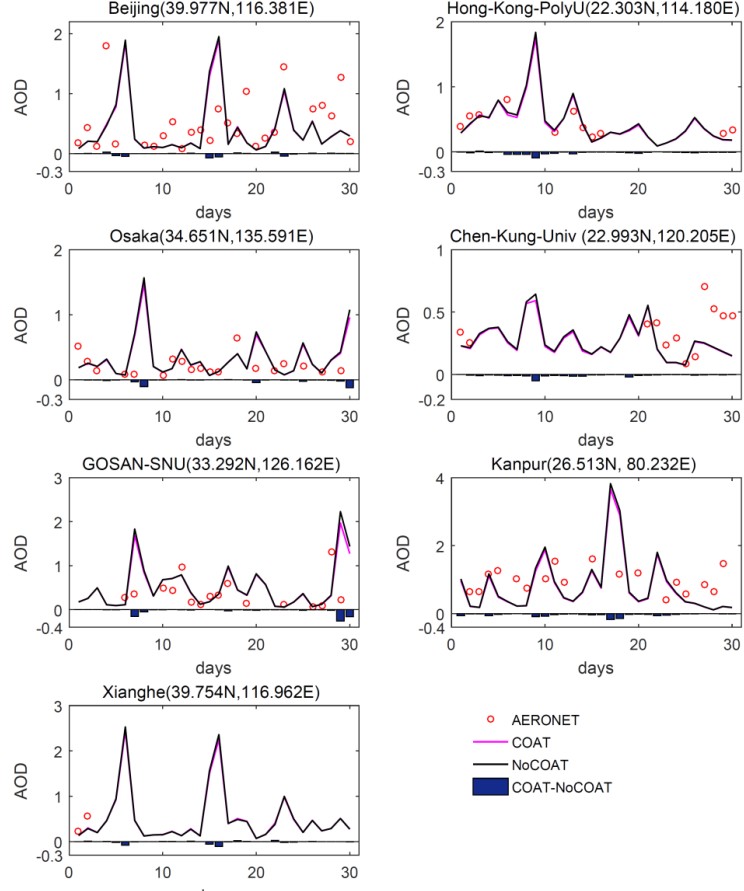

Fig.5. Time series of the modeled AOD from two experiments and their comparisons with AERONET measurements at 7 AERONET sites over East Asia.





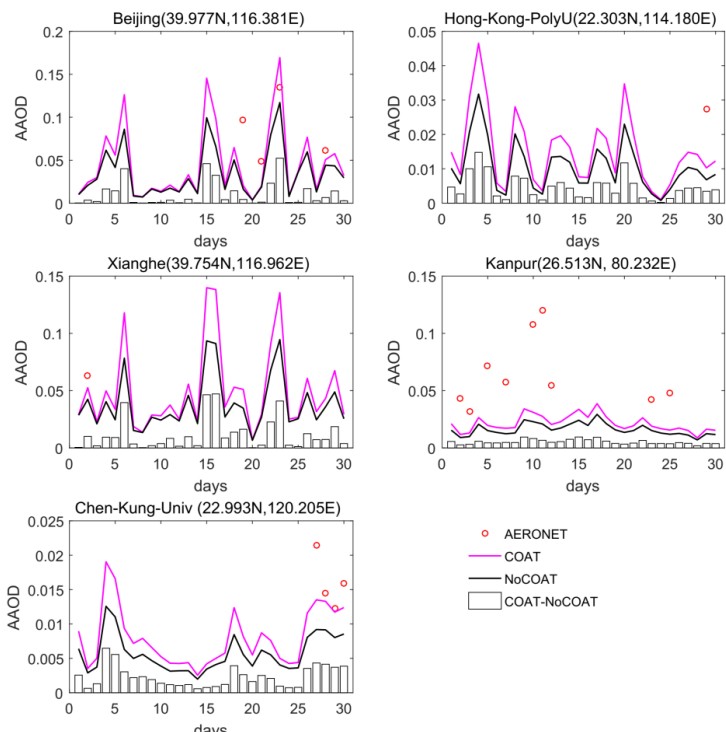

Fig. 6. Time series of the modeled AAOD from two experiments and their comparisons with AERONET measurements at 5 AERONET sites over East Asia.

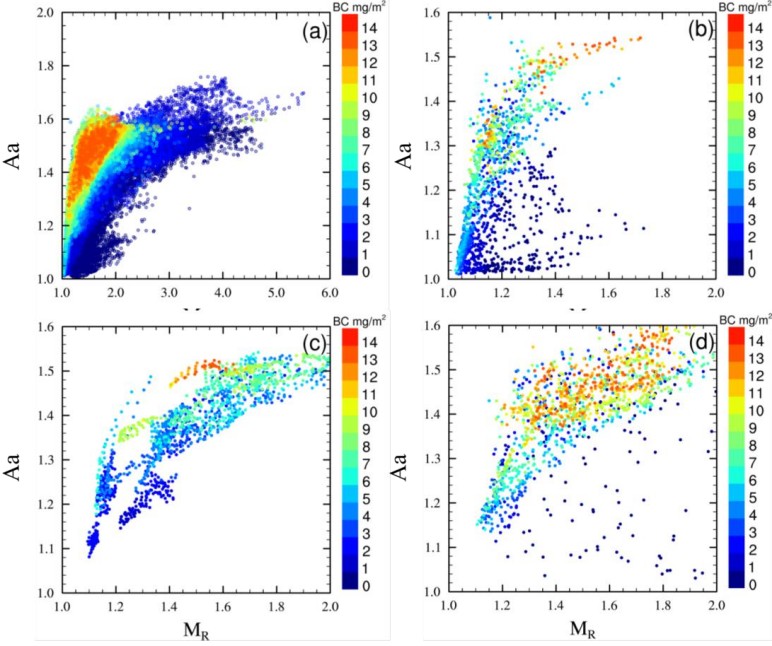

Fig. 7. Absorption amplification (Aa) and its dependency on $M_R$ and BC core over EC (a), Beijing (b), Nanjing (c), and Chengdu (d).





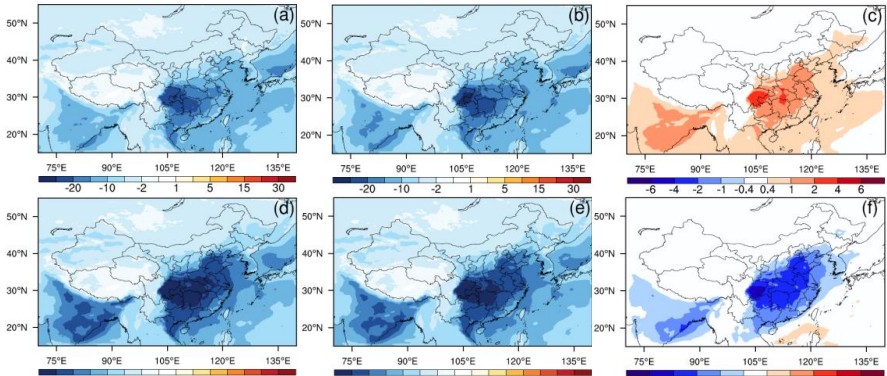

Fig.8. DRF under clear sky at TOA (top column) and SRF (bottom column). The DRFs shown from left to right are the results from COAT (a,d), NoCoat (b,e), and their differences (c,f).

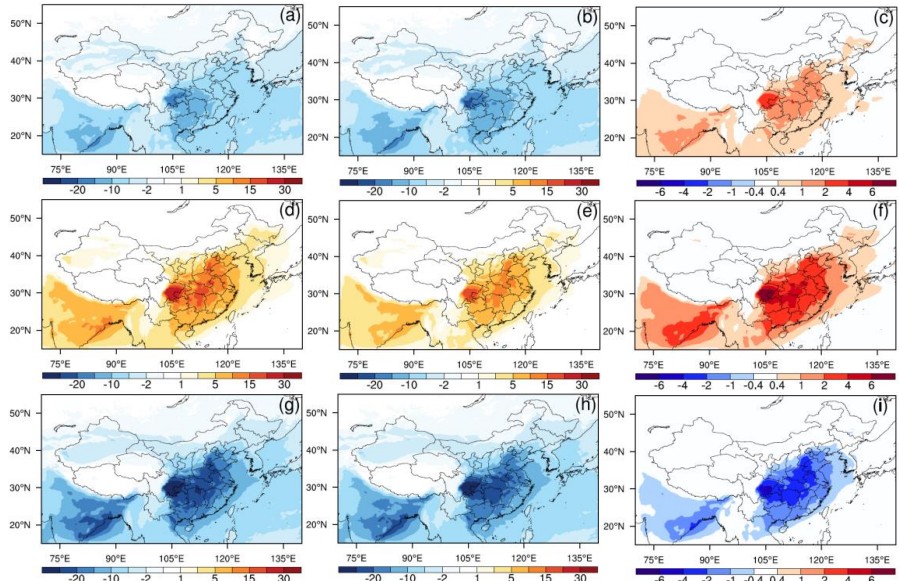


Fig.9. DRF under all sky, TOA, ATM, and SRF from top to bottom. The DRFs shown from left to right are the results from COAT (a,d,g), NoCoat (b,e,h), and their differences (c,f,i).





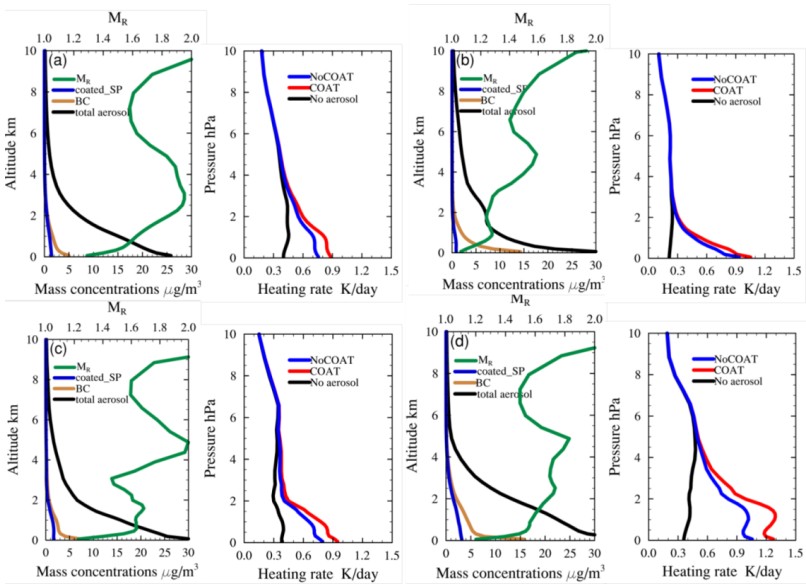

Fig.10. Mass concentrations of total aerosol, BC core, coated-SP, and $M_R$ are shown in the left, while heating rate from the experiments COAT, NoCOAT, and no aerosol in the right, over ECC (a), Beijing (b), Nanjing (c), and Chengdu (d).





Table 1. The monthly mean BC core mass, coated SS mass, the ratio of total mass (BC core + coated SS) to the BC
core mass ($M_R$), optical depth (AOD), absorption AOD (AAOD), direct radiative forcing at TOA and surface under
clear sky and all sky over East Asia (EA) and East-Central China (ECC) from the experiment COAT and NoCOAT (in
brackets).

|  | EA |  | ECC |  |
|---|---|---|---|---|
| BC core mass (mg m$^{-2}$) | 1.07 | (1.07) | 5.00 | (5.00) |
| Coated SS mass (mg m$^{-2}$) | 0.39 | (NA) | 1.45 | (NA) |
| $M_R$ | 1.36 | (NA) | 1.29 | (NA) |
| AOD | 0.36 | (0.37) | 0.86 | (0.88) |
| AAOD | 0.009 | (0.006) | 0.037 | (0.026) |
| Clear sky DRF$_{TOA}$ (W m$^{-2}$) | -6.76 | (-7.18) | -13.89 | (-15.02) |
| Clear sky DRF$_{SRF}$ (W m$^{-2}$) | -9.35 | (-9.02) | -24.53 | (-22.66) |
| All sky   DRF$_{TOA}$ (W m$^{-2}$) | -3.98 | (-4.32) | -8.34 | ( -9.35) |
| All sky   DRF$_{ATM}$ (W m$^{-2}$) | 2.43 | (1.76) | 9.49 | ( 6.84) |
| All sky   DRF$_{SRF}$ (W m$^{-2}$) | -6.41 | (-6.08) | -17.83 | (-16.19) |

Table 2. The statistics (mean and standard deviation) of the modeled AAOD from two experiments (COAT and
NoCOAT).

|  | COAT |  | NoCOAT |  |
|---|---|---|---|---|
|  | mean | std | mean | std |
| Beijing | 0.046 | 0.044 | 0.036 | 0.030 |
| Hong-Kong-PolyU | 0.015 | 0.011 | 0.011 | 0.007 |
| Xianghe | 0.050 | 0.037 | 0.038 | 0.024 |
| Kanpur | 0.021 | 0.007 | 0.016 | 0.005 |
| Chen-Kung-Univ | 0.008 | 0.004 | 0.006 | 0.003 |