# Peer review of "Mixing state of black carbon and its impact on optical properties and radiative forcing over East Asia"

_Atmospheric Chemistry and Physics, 2020_

## Referee Comment (RC1) · Anonymous Referee #1 · 14 Apr 2020

General comments:

This manuscript presents that by employing a nested GEOS-Chem-APM with predicted black carbon (BC) mixing state, the authors examined the effect of mixing state on aerosol optical properties, radiative forcing, and heating rate over East Asia. The manuscript is well written, and the structure is clear. Indeed, understanding aerosol optical properties and radiative forcing will improve future predictions of aerosol climatic effect. However, there are two fundamental issues in this paper: 1. The most highlighted findings have already been presented in the authors' previous papers. The model development, black carbon mixing states and aerosol optical properties have

been demonstrated in Yu et al. (2012) and Ma et al. (2012). For example, absorptive aerosol optical depth (AAOD) in the core-shell mixing experiment higher than the external mixing experiment has been shown in Yu et al. (2012). Many papers have studied black carbon mixing states and aerosol optical properties in regional or global models, or specifically in the geographic domain of East Asia. This manuscript does not address a different scientific question or give specific findings in their study compared to the previous works. For example, Grandey et al. (2018) have clearly shown that the representation of aerosol mixing state, size distribution and optical properties are the main causes of uncertainty in the strength of the cooling effect by exploring the representation of aerosols in a global climate model. They also presented similar conclusions regarding the aerosol direct radiative forcing, and the heating rate differences owing to black carbon mixing states. As to the geographic domain, East Asia, Zhuang et al. (2013), Sha et al. (2019), and many other papers have studied the aerosol mixing state and its radiative properties over China. Stevens and Dastoor (2019) even have one review paper on this topic. The authors need to do more literature review to highlight the unique findings in their work. 2. If the authors want to highlight the tool/model used in this study and quantify the mixing state of black carbon and its impact on optical properties and radiative forcing over East Asia, their numerical experiment is not sufficient, and the result is not statistically significant. For example, the monthly mean value in Table 1 should not just come from a one-month simulation in January 2014 with daily output. The comparison against observations in Figure 6 shows a lack of observational data during the study period. I think it could be improved if the authors expand their simulation to one year. A seasonal comparison would add value to this study owing to emission partition differences between BC and secondary species during each season. If computational cost is high, one month per season will provide a similar conclusion.

Based on the above reasons and considering that the authors need more time to rerun simulations and analyze data, I would like to reject this manuscript but encourage a resubmission after revisions.

Specific comments:

Lines 71-73: Many previous studies have discussed the core-shell mixing state of BC and its radiative properties and climate impacts. More literature review is needed. Line 80: Not necessary because there is no Section 2.2. Line 102: What is "CTM"? Line 107: Why choose January 2014? Line 112: What is "MIX"? Full name is needed. Lines 124-125: Reference? Why is it due to corn draw burning? Lines 166: Should explain how AAOD is calculated. Lines 179-180: They should be the modeled "daily" mean AOD and the AERONET "daily" results. Line 191: Should give a clear definition of absorption amplification (Aa). Lines 193-194: Abbreviation of the city seems not necessary. There are too many abbreviations in the paper. A list of abbreviations might be needed. Table 1: Should provide standard deviation as well.

Reference:

Ma, X., Yu, F., and Luo, G.: Aerosol direct radiative forcing based on GEOS-Chem-APM and uncertainties, Atmos. Chem. Phys., 12, 5563–5581, https://doi.org/10.5194/acp-12-5563-2012, 2012. Yu, F., Luo, G., and Ma, X.: Regional and global modeling of aerosol optical properties with a size, composition, and mixing state resolved particle microphysics model, Atmos. Chem. Phys., 12, 5719–5736, https://doi.org/10.5194/acp-12-5719-2012, 2012. Grandey, B. S., Rothenberg, D., Avramov, A., Jin, Q., Lee, H.-H., Liu, X., Lu, Z., Albani, S., and Wang, C.: Effective radiative forcing in the aerosol–climate model CAM5.3-MARC-ARG, Atmos. Chem. Phys., 18, 15783–15810, https://doi.org/10.5194/acp-18-15783-2018, 2018. Zhuang, B. L., Li, S., Wang, T. J., Deng, J. J., Xie, M., Yin, C. Q., and Zhu, J. L.: Direct radiative forcing and climate effects of anthropogenic aerosols with different mixing states over China, Atmospheric Environment, 79, 349-361, https://doi.org/10.1016/j.atmosenv.2013.07.004, 2013. Sha, T., Ma, X., Jia, H., Tian, R., Chang, Y., Cao, F., and Zhang, Y.: Aerosol chemical component: Simulations with WRF-Chem and comparison with observations in Nanjing, Atmospheric Environment, 218, 116982, https://doi.org/10.1016/j.atmosenv.2019.116982, 2019. Stevens,

[Figure]

R.; Dastoor, A.: A Review of the Representation of Aerosol Mixing State in Atmospheric Models. Atmosphere, 10, 168, 2019.

---

## Referee Comment (RC2) · Anonymous Referee #2 · 6 May 2020

This paper presents a modeling study for the East Asia region where the impact of BC mixing state on direct radiative forcing and heating rates is explored for the month of January 2014. The model system GEOS-Chem-APM is used for this study.

While the general topic is of interest to the community and within the scope of ACP, my major comment on this paper is that it is not clear to me what new insight is gained from this study. The GEOS-Chem-APM model system is a powerful framework for simulating aerosols, but we have known for a long time now that BC DRF is underestimated when BC is treated externally mixed, and core-shell treatment enhances absorption, which is one of the main outcomes of this study. A comparison of three cities is presented,

where differences in coatings and heating rates exist, but no process analysis is provided that would explain these findings. Over the past decade, a lot of work has been done in the area of black carbon radiative forcing and how BC mixing state modulates this, with many papers for the East Asia region and with mixing-state-aware models, for example by Oshima and co-workers and Matsui and co-workers. Those studies are not cited in the paper under review, so it is not clear if the study agrees or disagrees with prior work or what the new findings are. At the end of the conclusions, the authors outline some very interesting questions (how mixing state modulates boundary layer height, haze episodes). Pursuing those with this modeling tool would make a much more interesting paper with new insights for the community.

Specific comments:

1. Section 2.1: Since this paper is about BC mixing state, more detail needs to be provided how the APM model represents this. (What is the bin structure? Is coagulation included? Etc.) Even though this was published in previous work, this information is key for the reader.

2 How exactly was the NoCOAT simulation set up? Was the condensation process "switched off"? This would imply that the total mass concentrations in the NoCOAT and COAT simulations are different. Alternatively, is the aerosol material that would condense in the COAT simulation put into separate particles? Are the size distributions the same in the NoCOAT and COAT simulations?

3. Other missing information in this section are: What is the mixing state of aerosol that is transported into the domain through the boundaries? What is the model timestep? Why was January 2014 chosen for this paper? It seems arbitrary. Was there a spin up simulation time?

4. Section 3: A very much overlooked issue is the question about what is the mixing state that is assumed for carbonaceous aerosol at emission? The modeling approach for predicting mixing state evolution can be very sophisticated, but this won't help if the

emissions are not treated adequately.

5. Is there a temporal profile for the emissions (hourly, daily?)

6. Section 4: The sentence at the start of section 4.1 is not clear (line 128-130). Are you referring to mass concentrations or number concentrations?

7. Line 148: Unless the model includes the change in morphology during the first stage of aging, the fact that BC aging has two stages seems irrelevant here.

8. Line 161: Shell diameter, should read shell thickness.

9. Line 161: The refractive indices: This is an important parameter, I recommend listing these values explicitly. In particular, which species are assumed to be absorbing, just BC or also OC (brown carbon)? Stier et al. (2007) that uncertainties in BC refractive index are a leading cause of differences in predicted DRF, before even starting to think about mixing state.

10 Line 176: Suggest marking the sites in Figure 4. Include error metrics that quantify the agreement of simulated AOD and AERONET data.

11. Figure 2: the font of the wind arrow scale is too small. Mention in the caption that this is a monthly average. "Top column" and "bottom column" should read "top row" and "bottom row". "mass of BC core" should read "mass concentration of BC core"

12. Figure 4: Is this the absolute difference?

13. The English needs to be improved throughout the manuscript.

14. To improve readability I recommend refraining from using too many abbreviations. For example, the geographic locations (ECC, YRD, PRD, SCB, CC) and species group abbreviations (SS, SP, SNA) can easily spelled out without much effort.